

# Temperature profiles combined from lidar and airglow measurements

Thomas Trickl[1], Hannes Vogelmann[1], Michael Bittner[2,5], Gerald Nedoluha[3], Carsten Schmidt[2], Wolfgang Steinbrecht[4] and Sabine Wüst[2]

[1]Karlsruher Institut für Technologie, Institut für Meteorologie und Klimaforschung (IMK-IFU), Kreuzeckbahnstr. 19, 82467 Garmisch-Partenkirchen, Germany

[2]Deutsches Zentrum für Luft und Raumfahrt (DLR), Deutsches Fernerkundungsdatenzentrum (DFD), 82234 Oberpfaffenhofen, Germany

[3]Naval Research Laboratory, 4555 Overlook Avenue, SW, Washington, D.C. 20375, U.S.A.

[4]Deutscher Wetterdienst, Meteorologisches Observatorium Hohenpeißenberg, Albin-Schwaiger-Weg 10, 82383 Hohenpeißenberg, Germany

[5]Universität Augsburg, Institut für Physik, 86159 Augsburg, Germany

*Correspondence to:* Dr. Thomas Trickl, thomas@trickl.de, Thomas-Knorr-Str. 47, D-82467 Garmisch-Partenkirchen, Germany; tel. +49-8821-50283; Dr. Hannes Vogelmann, hannes.vogelmann@kit.edu, Karlsruher Institut für Technologie, IMK-IFU, Kreuzeckbahnstr. 19, D-82467 Garmisch-Partenkirchen, Germany; tel.: +49-8821-258

**Abstract.** In this study we examine the performance of the 354.8-nm Rayleigh temperature channel of the Raman lidar at the Schneefernerhaus high-altitude research station (UFS) in the Bavarian Alps (at 2675 m a.s.l.). The temperature reference value of the retrieval is adjusted to match the temperature determined from the OH* airglow around 86 km by the GRIPS instruments at UFS. In this way the quality of the 1-h measurements of the lidar is improved above 70 km. Comparisons were made between the UFS lidar, the MLS (Microwave Limb Sounder) satellite-borne instrument and the 354.8-nm temperature channel of Hohenpeißenberg (MOHp) differential-absorption ozone lidar. Between 35 km and 70 km we see a positive offset of the UFS temperatures with respect to the MLS values of up to about 9 K. This behaviour just slightly exceeds the expectations from earlier work. Despite a horizontal distance of just 40 km between UFS and MOHp acceptable agreement below 70 km was found in several cases. However, in general, the MOHp temperatures were slightly lower than those above UFS. We discuss potential technical issues and suggest solutions for upgrading the UFS lidar system. A significant enhancement of the laser repetition rate is recommended.

*Key words:* Temperature, stratosphere, mesopause, lidar, MLS, OH, airglow

## 1. Introduction

Accurate temperature measurements at short time intervals up to the mesosphere are an important contribution to climate research and for a better understanding of the atmospheric energy balance, in particular with respect to gravity waves. This can yield information for improving atmospheric and climate models that usually only consider the comparatively small-scale gravity waves in the form of parameterizations (e.g. Kim et al., 2003; Medvedev et al., 2019). These are simplified mathematical formulations of the wave effects, as exact calculations are hardly possible for resource reasons.

In the lower atmosphere the vertical distribution of the temperature is mostly obtained from the routine radiosonde ascents that are typically carried out up to roughly 30 km twice per day. Routine measurements up to higher altitudes are to a major extent provided by satellite-based sensors yielding temperature profiles up to more



than 90 km (e.g., Curtis et al., 1974; Russell et al., 1999; Waters et al., 2006;). These measurements are carried
out with a rather dense global coverage.
These observations are complemented by ground-based lidar measurements. Leblanc et al. (1998) stated that
lidar measurements provide the best vertical resolution and accuracy for middle-atmosphere temperature studies.
The most commonly applied lidar method is based on Rayleigh scattering (e.g., Hauchecorne and Chanin, 1980;
Hauchecorne et al., 1991; Fishbein et al., 1996; von Zahn et al., 2000; Keckhut, 2004) and yields temperatures
typically in the mostly aerosol-free range between 30 and 85 to 95 km. The temperature inversion from Rayleigh
backscatter profiles is the key approach within the global Network for the Detection of Atmospheric
Composition Change (NDACC, https://ndacc.larc.nasa.gov). Also, daytime measurements, based on extremely
narrowband spectral filtering and a very small field of view, have been demonstrated (e.g., Gille et al., 1991; von
Zahn et al., 2000).
The measurements times can be excessive and must be reduced to capture the short-term temperature variability.
This is achieved by the use of a powerful laser and a large receiver. Sica et al. (1995) have used a 2.65-m-
diameter spinning liquid-mercury mirror in order to collect a maximum of 532-nm light backscattered from 12
W from a frequency-doubled Nd:YAG laser and, in this way, have achieved temperature measurements up to
more than 100 km within 6 h of lidar operation and with vertical bin sizes of 48 m. At lower altitudes, shorter
measurement times are possible. The propagation of tidal and gravity waves was visualized in the altitude range
between 33 and 50 km (Sica, 1999).
Traditionally, backward retrievals starting at the highest useful altitude are applied (Hauchecorne and Chanin,
1980). This approach converges to the correct solution within roughly 15 km. In order to obtain more
quantitative data also in the uppermost range forward retrievals based on optimum-estimation methods have
been successfully applied (Khanna et al., 2012).
Above 80 km fluorescence lidar methods are used both for density and temperature measurements, up to 110 km.
The sensitivity of laser-induced fluorescence methods is very high and has made possible single-atom detection
in the laboratory (e.g., Ott, 2016). In the atmosphere its application is limited to low-density conditions where
fluorescence quenching is avoided. Metallic species, in particular sodium, potassium and iron atoms, have been
successfully detected and used for temperature retrievals (e.g., Bowman et al., 1969; Blamont et al., 1972; Mégie
et al., 1978; Fricke and von Zahn, 1985; Granier et al., 1989; She et al., 1990; Papen et al., 1995; Arnold and
She, 2003; Xu et al., 2006). This method has allowed for daytime operation (e.g., Chen et al., 1996; Chu et al.,
2001; Höffner and Lautenbach, 2009)
A combination of all the three lidar methods (rotational Raman scattering (e.g., Arshinov et al., 1983; Behrendt,
2005; Serikov and Bobrinikov, 2009), fluorescence, and Rayleigh scattering) has been applied to generate a
composite temperature profile from 1 km to 105 km (Alpers et al., 2004). This approach has been extended to
daytime operation by narrowband spectral filtering (Gerding et al., 2010; Kopp et al., 2015; see also von Zahn et
al., 2000). The propagation of waves could be studied in an enormous range between 35 km and 104 km.
Alternatively, temperature information has been obtained from ground-based measurements of the OH* airglow
(e.g., Sivjee, 1992; Scheer et al., 1994; Bittner et al., 2000; 2002; Beig et al., 2003; Reisin et al., 2014; Wüst et
al, 2023). This method is not height resolved, but benefits from the rather confined range of the OH* airglow
with its peak emission height between 85 and 87 km and a full width at half maximum (FWHM) between 6 and
9 km on average for the Alpine region (Wüst et al., 2017). Peak emission height and FWHM show an annual and
a semi-annual variation due to vertical transport processes at this height. Latitudinal and longitudinal variations
exist to a small extent (Wüst et al., 2020). Those values are retrieved from multi-year satellite data. From



ground-based OH* airglow measurements some height information can be deduced based on the altitude-
dependence of the vibrational population (e.g., Lopez-Moreno et al., 1987; von Savigny et al., 2012; Schmidt et
al., 2018, Noll et al., 2023)) or from the relation between the column-integrated volume emission rate and peak
emission height (e.g., Liu and Shepherd, 2006; Mulligan et al., 1995). The second method is based on the
functional relationship between both parameters; this must first be determined, e.g., by adding height-resolved
satellite measurements. In addition, the OH* layer has frequently been observed in limb geometry by rocket- and
satellite-borne spectrometers or radiometers (e.g., Baker and Stair, 1988; Takahashi et al., 1996; Englert et al.,
2010; von Savigny et al., 2013; Noll et al., 2017; Wüst et al., 2017; Li et al, 2021). Von Zahn et al. (1987), She
and Lowe (1998) and Schmidt et al. (2018) compared temperature measurements of sodium lidar systems with
results from OH* spectrometers. It should be noted that ground-state ($v'' = 0$) OH molecules have been observed
by laser-induced ultraviolet fluorescence, i.e. in the A $^2E^+$ - X $^2$II(0, 0) band at heights between 75-85 km
(Brinksma et al., 1998). The chemiluminescent OH*-layer addressed by passive remote sensing instruments
consists of pure rotational-vibrational transitions ($v'' \leq 9$) within the X $^2$II electronic state and, originating from a
different chemical reservoir, and peaks at higher altitudes (see Baker and Stair, 1988; Wüst et al., 2017).
In 2018 measurements of the atmospheric temperature in the stratosphere and the mesosphere were started with
the 355-nm channel of the big Raman-Rayleigh lidar system at the Schneefernerhaus Alpine high-altitude station
(UFS, 2675 m a.s.l.) in the Bavarian Alps (Klanner et al., 2021). These activities extend the water-vapour
(Vogelmann and Trickl, 2008) and aerosol (Trickl et al., 2024) sounding at this site. The backscatter profiles,
acquired within one hour, currently cover a range up to roughly 90 km. The inversion of these profiles yields
temperature data up to more than 80 km if a reliable upper-end temperature reference is available. At UFS we
benefit from using the temperature data from the OH* measurements at the station as a reference. This is a clear
advantage with respect to comparisons with satellite data, obtained on somewhat remote orbits, or with models
of unknown reliability (see Klanner et al., 2021).
In this paper, we aim at validating this new instrument, a task mandatory because of the features of the large
detection system and in view of the panned archiving in the data base of the Network for the Detection of
Atmospheric Composition Change (NDACC, https://www.ndacc.org). We look for the principal performance
and, thus, just give comparisons with a limited number of instruments carefully characterized elsewhere (for
references see below). In particular, we judge the performance by nearby lidar measurements and derive
recommendations for improvements and further tests.
**2. Data sources**
**2.1 Rayleigh backscatter measurements at 354.8 nm with a large lidar system at the Schneefernerhaus**
**high-altitude station (Zugspitze, Germany)**
Since 2009 a high-power Raman lidar system has been developed by a team from the Institute for Atmospheric
Environmental Research (Garmisch-Partenkirchen, Germany; since 2009 Karlsruher Institute of Technology,
IMK-IFU) that yields profiles of the water-vapour mixing ratio up to more than 20 km within one hour (Klanner
et al., 2021). This lidar is located at the high-altitude station Schneefernerhaus (Umweltforschungsstation
Schneefernerhaus, UFS, 47° 25′ 00″ N, 10° 58′ 46″ E, 2675 m a.s.l.) on the south side of Mt. Zugspitze (2962 m
a.s.l.), about 9 km to the south-west of IMK-IFU. Because of its elevation UFS provides mostly free-
tropospheric conditions (Carnuth et al., 2002). As pointed out by Trickl et al. (2024) the high base elevation
yields a significant gain in backscatter signal even in the stratosphere. The system is operated side by side with a
differential-absorption water-vapour lidar (DIAL; Vogelmann and Trickl, 2008; Trickl et al., 2014; 2015; 2016)



which allows for optimum calibration, also considering an absence of a significant bias, at least within the first
kilometres (Vogelmann et al., 2011; Trickl et al., 2016).
The Raman lidar system benefits from a large Newtonian receiver with 1.5 m diameter and a XeCl laser with up
to 180 W emission in single-line operation at a repetition rate of 350 Hz (Coherent, Lambda SX, with an
extended cavity containing an intracavity etalon and a polarizer).
This system can additionally yield temperature measurements by two methods. Up to the lower stratosphere the
temperature may be retrieved from two rotational Raman channels. This approach has been successfully tested
(Höveler, 2015), but has not yet entered routine operation because the focus has so far been on optimizing the
performance of the $H_2O$ channel.
The second approach is inverting one of the two Rayleigh backscatter profiles of the lidar at 307.96 nm and
353.14 nm. The emission for the preferred second channel was first generated by stimulated Raman shifting the
output of the XeCl laser in hydrogen (Klanner, 2022). However, the performance at 353.14 nm was not reliable
at the high repetition rate of 300 Hz. As a consequence we added a frequency-tripled injection-seeded Nd:YAG
laser (Continuum, Powerlite Precision 8020) previously used in our water-vapour DIAL. This laser emits at the
wavelength of 354.81233 nm and yields pulse energies of 170 mJ, substantially more than the emission planned
from the Raman-shifted excimer laser. However, the repetition rate is limited to 20 Hz. The choice of the
ultraviolet wavelength provides the advantages of a larger backscatter signal than for 532.2 nm. Most
importantly, the light-absorption by ozone at 532.2 nm is avoided (Trickl et al., 2024).
A general design feature in our lidar systems is the absence of optical fibres. In this way, coupling losses, near-
field issues and fluorescence effects are avoided. In addition, the polarization of the transmitted radiation is
conserved which is essential for the chosen optical configuration of the detection polychromator (Fig. 8 of
Klanner et al., 2021). The 354.8-nm channel is spectrally filtered by a combination of a filter (Semrock),
transmitting 85 % – 90 % of the radiation between 330 and 355 nm and blocking spectral contributions outside
this range by at least $10^5$, and a narrowband filter with a width of less than 1.2 nm (Alluxa).
A key issue for temperature measurements is a high quality of the detection electronics: An uncertainty of (e.g.)
the order of 1 K means a relative uncertainty of the measurements of less than 0.5 %. Here, we benefit from the
experience gained from our older systems (e.g., Trickl et al., 2020; 2023; 2024). The backscattered radiation is
detected with Hamamatsu R7400U-03 photomultiplier tubes (PMTs) with actively stabilized sockets from
Romanski sensors (RSV). In the following we describe, for simplicity, the signal voltages as positive values
instead of the negative ones emerging from the detectors. The performance of these detectors is highly linear for
peak analogues signals cautiously set to 70 mV and less (Trickl et al., 2020; 2024). Here, this is achieved by
adjusting the laser pointing in order to maintain the maximum far-field signal level. However, the analogue
output is ten times more sensitive to the background signal than the PMTs from the older 5400 series (Trickl et
al., 2020). Undershooting occurs at analogue background levels exceeding 1 mV (for 50 Ω termination), which
does not matter during night-time. However, also excessive lidar backscatter could, in principle, lead to some
undershooting and we have, therefore, limited the peak analogue signal even to less than the 70 mV (into 50 Ω)
successfully applied by Trickl et al. (2020).
The output is registered in 12-bit transient digitizers at a range-bin size of 7.5 m (Licel). This model was
modified in 2010 for our ozone DIAL with a ground-free input, which was also implemented in all systems
digitizer systems purchased in the following. The Licel transient digitizer is optimized for very low noise (less
than $\pm 10^{-5}$ of the measurement range for 4096 laser shots), but, in contrast to the previously used system from
DSP Technology (model DSP 2020; Trickl et al., 2020), features a slight exponential wing that grows with the



area of the lidar backscatter signal. This wing could be easily corrected in the case of the ozone DIAL because of
the short decay time of the signal (Trickl et al., 2020), but is difficult to quantify for the long signal at longer
wavelengths. An exponential curve is subtracted, derived from a comparison with the photon-counting results at
high altitudes. The decay constant is the same for all retrievals.
Parallel to this detection channel for the analogue signal we count the individual photons in a MCS6a
multichannel scaler (FAST ComTec) with a pulse-pair resolution of 0.2 ns, more than sufficient for the 1.5-ns-
wide pulses from the detector. Because of the near-field overload of the sequential data transfer to the computer
we delay the start of the counting system by 10 µs. The perfect linearity of the photon-counting channels has
been verified by the validation of our water-vapour measurements (Klanner et al., 2021) and the performance of
our stratospheric aerosol measurements (Trickl et al., 2024), at least to up to 45 km. We have been optimistic
that the perfect linearity extends to much higher altitudes since the settling time of artefacts for this type of PMT
seems to be rather short. For the threshold voltage of –4 mV set at the fast discriminator (RSV) testing with
closed detector entrance revealed that the detection is free of "dark counts". For the temperature measurements,
the night-time background count rate is of the order of 0.2 counts per 7.5-ns bin and hour at the maximum
distance of 120 km, which allows for keeping the far-field signal-to-noise ratio as high as possible.
**2.2 Microwave Limb Sounder on board the Aura satellite**
The Microwave Limb Sounder (MLS; Waters et al., 2006) on board the Aura satellite (Schoeberl et al., 2006)
was launched to space in 2003. The instrument not only detects thermal emission from various molecules (most
recently, e.g., Nedoluha et al., 2024; 2025), but also yields vertical profiles of the atmospheric temperature up to
almost 100 km. 3500 vertical profiles are produced per day. The temperature retrieval is based on limb radiances
near the oxygen spectral lines at 118 GHz and 234 GHz. The limb-tangent pressure is derived from the shape of
the broadened $O_2$ lines.
Schwartz et al. (2008) analysed the quality of the temperature and geopotential height determined from the
measurements. Comparisons with other satellite-borne instruments indicate a negative temperature bias for
altitudes corresponding to pressures of 1 mbar and less. Mevedewa et al. (2014) report a negative bias of about
–10 K of the MLS temperature with respect to that obtained from OH measurements above 80 km.
For the examples shown in this paper there are just two of the 14 cases with a passage almost perfectly above
Garmisch-Partenkirchen during the night-time lidar observations. Given the enhanced wave activity that can be
expected above the Alps (e.g., Hannawald et al., 2019) this limits the quality of comparisons.
The geopotential height is converted to absolute height for the comparisons.
**2.3 GRIPS OH\* chemiluminescence**
A closer spatial agreement with the UFS lidar is achieved with the routine OH\* airglow measurements at UFS in
the mesopause region. The OH spectra are registered with the Ground-based Infrared P-branch Spectrometer
(GRIPS) instrument (Schmidt et al., 2013), integrated in the global Network for the Detection of Mesosphere
Change (NDMC; https://ndmc.dlr.de; e.g., Reisin et al., 2014). The centroid altitude of the airglow is 86 ± 2 km
and slightly varies with the season (von Zahn et al., 1987; Baker and Stair, 1988; She and Lowe, 1998; Wüst et
al., 2017, 2020). The aims of the research are an identification and quantification of the influence of climate
change, the detection of solar activity effects, the influence of atmospheric dynamics (wave activity) and the
validation of satellite instruments.





A survey on the relevant spectroscopic data and literature is given, e.g., by Noll et al. (2020) and Wüst et al.
(2023). Wüst et al. (2023) provide a review on the recent achievements of OH* measurements. Only transitions
emerging from low-lying vibrational levels appear to be suitable for sufficiently quantitative temperature
derivations and we, therefore, use the low-lying (3,1) OH* transitions (Bittner et al., 2002; Noll et al., 2017). The
lack of vertical resolution and the relatively large thickness of the OH* layer yields temperatures that are
weighted means over a layer of the atmosphere extending over roughly 8 km.
The GRIPS instrument at UFS is oriented for slant-path detection which means that the OH* layer is reached
approximately above Bozen (46.6° N, 11.0° E, Italy) south of the Central Alps, at a horizontal distance of
roughly 90 km. The night-time temperature time series exhibit variations such as can be seen in the three
examples shown in Fig. 1. The amplitudes of the T variations can be as high as 10 K, rarely more. In order to
gain insight into the horizontal extent of these structures we, therefore, compared the GRIPS measurements with
those at Oberpfaffenhofen (OPN, 48.1°N, 11.3°E, about 77 km to the north) during the same nights. The full-
night average temperatures at both sites mostly agree within 1 to 3 K. The structures measured from the OPN
station are not identical, but at least similar (see right panels in Fig. 1). The cases with pronounced differences
demonstrate that temperature changes can occur on a confined spatial scale. This leads to uncertainties for the
comparisons in Sect. 3.
The temperature exhibits a pronounced seasonal dependence varying between about 170 K in summer and 220 K
in winter, with average amplitudes of 17.5 K and 3.0 K of the annual and semi-annual component (see Schmidt
et al., 2023). There is no consistent trend in the night-time series used this study, but a general tendency for a
slight rise. Systematic temperature changes throughout the night caused by the semidiurnal migrating (12 h) tide,
maximising in winter can be expected. On average this can lead to a 4-5 K temperature increase from 21 CET to
3 CET at UFS in January (compare Fig. 7 in Schmidt et al (2023)). As far as the nights used in this study show
variability at all, it appears to be dominated by shorter period structures, which in some cases exhibit substantial
amplitudes (compare Figure 2).

### 2.4 Temperature measurements with the Hohenpeißenberg ozone DIAL

A considerable chance for the validation of the UFS lidar is given by the routine night-time temperature
measurements with the ozone differential-absorption lidar (DIAL) at the Meteorological Observatory Hohen-
peißenberg (MOHp) of the German Weather Service. This station is located just 42 km to the north of UFS.
The MOHp DIAL is technically similar to the UFS lidar (Wing et al., 2021) which is an excellent basis for
comparisons. For the T measurements it is also operated at 354.8 nm using a frequency-tripled Nd:YAG system
(Innolas, model SpitLight 600, 120 mJ per pulse, 20 Hz repetition rate). The PMT type is similar (Hamamatsu,
R9880U-110, with home-made socket) and was compared with the PMT type used at UFS without showing a
difference in lidar backscatter profile. The diameter of the principal mirror of the Newtonian telescope is smaller
(1.0 m), but the data acquisition is extended to full nights, which results in a high signal-to noise ratio.
Exclusively photon counting is applied (FAST, P7882-2, maximum count rate 200 MHz).
The data are averaged over 150 m and, after the evaluation, smoothed to 450 m. The temperatures are routinely
stored in the NDACC data base (https://ww-air.larc.nasa.gov/missions/ndacc/data.html#) from 27 km to roughly
95 km.

### 3. Inversion of the UFS lidar data



**3.1 Temperature retrieval**
The temperature profiles are retrieved from the 354.8-nm backscatter profiles as described by Klanner et al.
(2021). Here, we give a few more details than in the preceding publication, based on the experience gained from
the more recent measurements.
The algorithm is a modified version of the downward inversion method introduced by Hauchecorne and Chanin
(1980; see also Shibata et al., 1986). This method has the advantage of self-correction within roughly 15 km
from the top (reference) altitude and is less complicated than the forward inversion proposed by Khanna et al.
(2012) which might be implemented at a later stage. However, we benefit from the OH* airglow measurements
at UFS that allow us to shift the reference temperature to a value that yields a good match of the T profile with
the OH temperature at 86 km, the average altitude of the OH* layer (Wüst et al., 2017). In cases in which the
reference altitude has to be chosen below 86 km an extrapolation is estimated. In cases in which the reference
altitude is above 86 km we used the temperature from the U.S. Standard Atmosphere (1976) in a first step and
apply a correction in a following retrieval that shifts the temperature at 86 km to a value within the uncertainty
range of the GRIPS temperature.
The main source of data is the photon-counting signal. We select a bin size of 51.2 ns, which represents a length
of 7.6747 m. The PMTs were tested without input radiation and turned out to be free of background counts for 1
h of data accumulation in a single bin. Thus, no cooling of the detector is required. As mentioned, in night-time
atmospheric measurements the average background count rate has been as low as roughly 0.2 counts bin$^{-1}$ h$^{-1}$
(Fig. 1). This is the result of careful spatial filtering and the narrow spectral filtering. This performance to a
major extent excludes artefacts such as those described by Wing et al. (2018a). The backscatter profiles from the
counting system are interpolated to match the 7.4948 m bin size and the vertical grid of the transient digitizer.
We start the data evaluation with visualizing the raw data and their conversion. In Fig. 1 we show an example of
backscatter profiles from 9 October 2021. After determining the zero point of the distance scale we average the
content of two neighbouring bins and double the size of a single data bin to 14.9896 m, which reduces the
number of bins to 8000 (119.917 km). The analogue data are substantially noisier than the photon-counting data
and exhibit a background drift tentatively ascribed to the performance of the transient digitizer. The 355-nm
analogue profile was corrected by subtracting $3.22\times10^{-4}$ exp($-8.0\times10^{-6}$ ×r), with r being the distance above the
lidar in metres. The constant of the exponential decay was used in all measurements evaluated since April 2021.
This decay is 7 times slower in comparison with that in our ozone DIAL (Trickl et al., 2020) and is observable
just because of the excessive operating range and the longer averaging. The fast decay component of the DIAL
(277 to 313 nm) cannot be resolved because of the slower decay of the 355-nm signal, also implying the absence
of absorption in ozone.
The analogue data are used from about 10 km to 30 km. As can be seen in Fig. 1 the correction is crucial for
obtaining optimum temperature profiles from the analogue data for distances r up to 30 km. The data correction
must be made with care because of the enormous requirements for the temperature measurements. Below 10 km
the signal is partially cut off by one side (blade) of the entrance aperture, the laser beam being adjusted to limit
the near-field peak signal. Above 30 km the photon-counting signal is taken where it is fully linear. No dead-
time correction has been implemented in this channel as frequently done in lidar systems based on counting
alone. All backscatter signals are carefully compared with simulations from meteorological data (see also
below).
The analogue measurement in Fig. 1 is the best in comparison with those in the other measurements obtained
since spring 2021. In other cases, slight deviations from the clean exponential behaviour were observed that we



tend to ascribe to interference of the magnetic field of the laser flashlamps as found in earlier work (e.g., Trickl ,
2010). The use of a diode-laser-pumped laser system is planned (see Sect. 4). For such a laser system we have
not found any similar interference (Trickl et al., 2024).
For reducing the data noise we apply altitude-depended smoothing with a numerical filter as described by Trickl
et al. (2020). The bin interval of the Blackman-type filter is nonlinearly enhanced using the formula
$a + i^2 \times b$
at bin i (size 14.9896 m), with normally a = 20 and b = $4 \times 10^{-5}$. This results in a variation of the VDI (1999)
vertical resolution from 58 m at the ground to 1216 m at 50 km and a maximum of 2908 m at and above 76.7 km
a.s.l. (Trickl et al., 2020; VDI, 1999; see (Iarlori et al., 2015; Leblanc et al., 2016) for other definitions). The
VDI resolution is defined as the range difference between 25 % and 75 % of the response to a Heaviside step.
Above this, the vertical resolution stays constant to limit the computation requirement.
As described by Klanner et al. (2021) we first calculate the atmospheric density in a fully quantitative Klett-type
approach with downward integration from the far end (Klett, 1981; 1985):
$$n(r) = \frac{n(r_{ref})\ r_{ref}^2 S(r_{ref})}{r^2 S(r) + 2n(r_{ref})\sigma_R \left[ \int\limits_{r}^{r_{ref}} r'^2 S(r')dr' \right]} \ , \tag{1}$$

S$(r)$ being the backscatter signal, $r_{ref}$ the reference distance and $\sigma_R$ the Rayleigh extinction coefficient. Here, the
misprint in Eq. 1 of Klanner et al. (2021) is corrected that has not been part of the computer program. The far-
end reference density $n(r_{ref})$ is varied for matching the density derived from the Munich radiosonde (launched at
Oberschleißheim, station 10868, about 100 km roughly to the north), also making comparisons with the densities
from NCEP (National Centers for Environment Prediction) at higher altitudes. We download the sonde data from
https://weather.uwyo.edu/ upperair/sounding.html. The NCEP altitudes, pressures and temperatures are taken
from the web site of the Network for the Detection of Atmospheric Composition change as calculated for
Garmisch-Partenkirchen (https://www-air.larc.nasa.gov/ missions/ndacc/data.html?NCEP=ncep-list). The NCEP
listings are available up to roughly 55 km.
From the density $n$ we obtain the temperature profile by applying
$$T(z) = T(z_0)\frac{n(z_0)}{n(z)} + \frac{m_{air}}{k\ n(z)} \int\limits_{z}^{z_0} n(z')g(z')dz' \ , \tag{2}$$

with $z$ being the altitude above sea level, $z_0$ the far-end reference altitude, $m_{air}$ = 28.9644 u (U.S. Standard
Atmosphere, 1976; 1 u = $1.66053904 \times 10^{-27}$ kg (Mohr et al., 2016)) the mass of an "average air molecule", and g
the gravitational acceleration,
$$g(z) = g_0 \left(\frac{r_E}{r_E + z}\right)^2 ,$$

with $g_0$ = 9.80665 m s$^{-1}$ and the earth radius $r_E$ = 6356766 m (U.S. Standard Atmosphere, 1976). In the U.S.
Standard Atmosphere the atmosphere is regarded as well mixed up to 86 km, which suggests to be cautious at
the highest altitudes accessible.
The inversion starts at the highest reasonable altitude, mostly below the upper end of the density profile.
Normally, we take the temperature for the start altitude from the U.S. Standard Atmosphere (1976) as the
reference. The algorithm in Eq. 2 is self-correcting: Within 15 km of downward integration the temperature is



does no longer vary as a function of the reference value by more than ±2 K (Klanner et al., 2021), as a
consequence of the density dependence of the first term of Eq. 2. It turned out that the influence of the reference
density (Eq. 1) is very low and can be neglected for the careful adjustment made (see above).
In eight of the fourteen cases examined we had to make adjustments of the reference temperatures to match the
temperature of the GRIPS instrument at 86 km, or, in one case without GRIPS measurements, the range of MLS
values. For a perfect comparison of lidar and GRIPS an extension of the operating range of the lidar to more than
100 km is desirable. Wing et al. (2018a) recommended to use the MSIS model output as a reference. The result
was not satisfactory at all (Klanner et al., 2021) and we decided to use the GRIPS temperatures.
The retrieval of temperature from lidar data is a highly demanding task. For instance, an uncertainty of 1 K
means a relative uncertainty of 0.33 % at a temperature of 300 K. Thus, a very high quality of the backscatter
signals is a prerequisite for reasonable results.
Aerosol corrections have been tested and lift the temperature in a range around 22 km, typically by 2 to 4 K
under the conditions of the cases chosen. The aerosol backscatter coefficients are taken from the stratospheric
aerosol measurements (Trickl et al., 2024) from adjacent days and converted from 532.24 nm to 354.8 nm by
multiplication with 1.992 (Jäger and Deshler, 2002; 2003). However, such an effort makes sense just if the
quality of the exponentially corrected analogue signal matches that in Fig. 1 which was the case just for two
measurements (16 November 2018 and 9 October 2021). In the other cases, the analogue signal background
exhibited slight modulations perhaps caused by the magnetic field of the flashlamp-pumped laser. Here, the
exponential correction is adjusted to generate temperature values matching those from the midnight Munich
radiosonde (station 10868).
**3.2 Uncertainties**
As pointed out in the U.S. Standard Atmosphere (1976) the atmospheric composition is homogeneous mixed up
to 86 km. Local short-term perturbations are averaged away by the long signal acquisition time of 1 h.
The uncertainty from shot noise, as derived from the smoothed signal, is of the order of 10 K at 70 km. We use a
scaling law
$$u = \sqrt{u_0^2 + (u_1 \frac{r^2}{r_{ref}^2})^2 + (u_2 S(r))^2} \ , \qquad (3)$$
for the uncertainties, with coefficients $u_0$, $u_1$, and $u_2$ that are estimated by comparison with reference
measurements at lower altitudes and the shot noise at higher altitude. The second term in Eq. 3, quadratic in r,
reflects the quadratic rise of the noise of the unsmoothed quantities. We select the reference distance $r_{ref}$ at the
upper end of the data evaluation range. By the approach with Eq. 3 considerable computation efforts have been
avoided. The results of the night-time series (Sect. 4.4) verify the approximation.
**4. Results**
**4.1 UFS lidar results**
The lidar test phase presented in Sects. 4.1 to 4.3 covers the 14 measurements made between November 2018
and February 2022. In addition, four separate night-time series are presented in Sect. 4.4.





In Fig. 3 we show the result of a revised data evaluation of the example shown by Klanner et al. (2021) based on
the slightly enhanced smoothing described in Sect. 3.1 and an exponential correction of the analogue profile that
was missing in the earlier evaluation. In addition, an aerosol correction was made that resulted in good
agreement with the temperature profile of the Munich radiosonde down to 12 km. The ripple below 32 km is a
result of the aerosol correction. Below 12 km the signal was partly cut off by rotating the laser beam to shift the
return bundle towards one side the entrance aperture of the detection system in order to limit the maximum
signal.
The GRIPS value was determined from an average of the GRIPS temperatures during the measurement period of
the lidar, which was done for all cases. The error bar represents the standard deviation for that period and does
not include the uncertainty due to the shifted horizontal position of the OH* layer south of the Central Alps. This
influences the results in an uncertainty of ±5 K, sometimes more. The GRIPS value in this example agreed
sufficiently well with the lidar temperature generated by initiating the retrieval with the U.S. Standard
temperature at 88 km. Thus, the calibration was not shifted.
The NCEP data agree well with the radiosonde temperatures up to the operating range of the balloon. There is a
small deviation with respect to the lidar at 40 and 46 km. We do not know if this bias is caused by a vertical shift
of the NCEP profile.
We also added temperature profiles from MLS and the MOHp lidar. Unfortunately, no T profile from MOHp
was available for the same night, and we took the results from the two neighbouring measurement nights. The
average of both MOHp profiles agrees quite well with the UFS profile to about 70 km. The MLS profiles are
shown for about 13:31 CET (Central European Time, three thin orange lines) and about 3:05 CET on 17
November 2018 (three blue lines), named "noon" and "midnight", respectively. The agreement should be better
for the "midnight" measurement, but the best agreement is limited to the spike region around 64 km. In principle,
this could be ascribed to the pronounced difference in longitude.

### 4.1 Comparisons of the temperatures from the UFS lidar and the NCEP archive

Up to the burst altitudes of the balloons (mostly 30 to 33 km) the temperature of the Munich radiosonde is nicely
reproduced. For higher altitudes the temperatures are compared for all NDACC stations from the NCEP data.
The NCEP profiles are listed up to roughly 55 km. Again, geopotential heights are converted to absolute heights.
Trickl et al. (2024) found for the evaluation of stratospheric aerosol that the Rayleigh reference profiles
calculated from the NCEP density and temperature data look highly reliable up to the chosen Klett (1983)
reference distance of 45 km (absolute altitude 47.675 km). However, the requirements for the temperature are
much higher. For the temperature data we found, in comparison with the UFS lidar, reasonable agreement up to
35 km, but sometimes deviations above this altitude (and below 50 km) of up to about 10 K occur (see Fig. 3 and
examples in the following sections; in one case even more). Sometimes there is the impression of a vertical shift
of the NCEP profile or a slightly wrong gradient in the range of the strongest temperature increase. The NCEP
profiles are calculated for noon UTC (13:00 CET) which certainly limits the comparability.

### 4.2 Comparison the temperatures from the UFS lidar and MLS

After initially concentrating just on the co-ordinates of the satellite for a given date (e.g., Klanner et al., 2021)
we now also looked at the times of the MLS overpasses, which resulted in a somewhat different view. The mean
passage times specified were 1.38 (±0.40) h with respect to noon (CET) and 2.69 (±0.39) h with respect to the



following midnight (CET). The "midnight" passages can be expected to be the better choice for the comparisons
with the early-night lidar measurements and were normally taken for the statistical evaluation. Morning times on
a given day were no longer used. For the downloaded data we consistently found the three latitudes 45.9º N,
47.35º N and 48.8º N which is sufficiently close to UFS (within ±1.6º). By contrast, the longitudes were not
reproducible and ranged between 3.2º E and 20.7º E. For just two of the days the longitude perfectly matched
that of UFS. However, these profiles did not agree with those from the lidar significantly better. In two cases one
measurement time group of MLS was missing, and we took the profile for a passage at a time difference of about
±12 h with respect to midnight.
In addition to Fig. 3 we show in Fig. 4 six examples of comparisons. On average the positive bias of the
temperatures from the lidar and that from MLS remained. The best agreement below 70 km is seen in the
midnight comparisons (MLS: thin blue lines) in Fig. 3 and Fig. 4e. In three of the cases (not shown in Fig. 4) big
positive temperature peaks exceeding the estimated uncertainties could be seen above 70 km in the lidar data.
One of these cases is shown in Sect. 4.3. In addition, in Fig. 4f there is a rather excessive deviation between lidar
and MLS at 69 km, but here the temperature difference with respect to the MOHp lidar is smaller (thick blue
line). Because of the good latitudinal matching this discrepancy is strange. We are unable to explain it. One
could think either about wave effects or about an outlier of the lidar measurement. For identifying waves one
needs time series which is possible because of the one-hour time resolution of the system.
A very strange deviation was untypically found at rather low altitude for 12 December 2018 (not shown). At the
moderate altitude of 46 km the midnight MLS temperature was lower by as much as 25 K than that from the
lidar. The difference with respect to the MLS noon temperatures had been just 15 K, also this, however, being
out of any expectation for this altitude range. Below 35 km and above 57 km the agreement was reasonable to
excellent. For our statistical analysis (see below) we took the noon profile.
In Fig. 5 we show the differences of the profiles for all measurements, as well as the arithmetic average and the
positive and negative standard deviation. For the averages we selected the MLS latitude 47.35º N, −0.07º away
from that of UFS. In all but two cases the "midnight" MLS profile was taken. The maximum deviations for the
lidar are 9.4 K at 52.8 km and 8.6 K at 68.6 km. The minimum difference is found at 61.0 km is 3.1 K.
The negative temperature deviation of MLS confirms the findings in earlier work. Schwartz et al. (2008)
compared the MLS temperature profiles with those from several other satellite-borne sensors. They abruptly
started to depart from the other profiles above 45 km, the maximum negative offset reaching values between 5
and 8 K. The minimum difference around 60 km is also confirmed. Wing et al. (2018b; 2021) found a similar
behaviour for other Rayleigh lidar systems.

### 4.3 Comparisons of the temperatures from the UFS and MOHp lidar systems

Instead of comparing the UFS results with another satellite-borne sensor such as the proven SABER (Sounding
of the Atmosphere using Broadband Emission Radiometry; launch: December 2001; Russell et al., 1999;
Mertens, 2009; Esplin et al., 2023) we preferred to compare our temperature results with those of the
nearby MOHp ozone lidar. The advantage is the mentioned relatively small distance between the two lidar sites
The MOHp temperatures have been demonstrated to match the SABER temperatures almost perfectly (Fig. 10 of
Wing et al., 2021), as is the case also for other lidar stations (e.g., Cooper, 2004; Wing et al., 2018b; Dawkins et
al., 2018). Thus, we see the MOHp lidar as a suitable reference (see also Steinbrecht et al., 2025). During 8 of
the 14 measurement nights at UFS the MOHp lidar was operated as well.



As expected from earlier work (e.g., Wing et al., 2018b; 2021) the UFS and MOHp lidar temperature channels
yield a better agreement than the comparisons of the UFS lidar with MLS. However, the temperatures for the
two lidar sites may differ. In Fig. 6 the best case is presented with almost identical values up to 71 km (8
November 2021). Above this altitude the UFS temperature profile exhibits a pronounced peak strongly outside
the estimated uncertainty above this altitude. The MLS values sometimes deviate strongly which is associated
with bad temporal and spatial co-incidence.
The excellent agreement of the lidar results at both stations in Fig. 6 suggests that there is no fundamental
mismatch of the retrieval algorithms. Also in Fig. 7 the results do not differ much, except for the range above 66
km. There, the deviation is, still, within the expected uncertainty or the temperature variations suggested by the
variability in the GRIPS data. This agreement also suggests the absence of fluorescence effects of the
interference filters or PMT issues (see Sect. 5).
However, the examples in Fig. 4 demonstrate that there is, on average, a systematically lower temperature in the
MOHp values. This is somewhat surprising because of the good matching of the MOHp temperatures with
SABER (Wing et al., 2021) and a similar average offset between SABER and MLS (Schwartz et al., 2008) as
found for the UFS lidar in Fig. 5. At least, the agreement between the two lidar systems is clearly better than that
in Fig. 5.
This behaviour will be discussed in Sect. 5.

### 4.4 Comparisons of night-time series at UFS with the MOHp all-night temperature profiles

One issue of comparisons of the lidar measurements at UFS and MOHp is that the measurements at UFS are
confined to one hour and were, until 2022, concentrated to the first half of the night, whereas the soundings at
MOHP cover full nights. Although on average a night-time temperature increase has been found at the altitude of
the OH layer and ascribed to atmospheric tides we speculated on potential night-time cooling of the atmosphere
below. Of course, such an effect is difficult to evaluate in the presence of pronounced atmospheric wave
activities.
We, therefore, carried out four night-time series in November 2024, i.e., after the end of the period of
investigation in the previous sections (February 2022). The measurements were carried out with durations of just
half an hour, which resulted in a higher variability of the values at high altitudes than for the 1-h measurements.
Figure 8 shows the examples for the last two of the four nights, for which MOHp reference profiles existed and
for which the GRIPS temperature variability was low. During these nights the differences of the single-night
average UFS and the MOHp temperatures are rather small. Very few profiles depart strongly from the average,
beyond our expectations.
In Fig. 9 we present temperature time series of these two nights for altitudes differing by 5 km between 30 and
80 km. Up to 55 km the series are sufficiently smooth to allow us to judge temperature trends. There is no
indication of major night-time atmospheric cooling over a large altitude range, which is favourable for a good
agreement of the two lidar systems. The negative temperature development at 45 km and 50 km during the night
between 4 and 5 November could be an exception, but does not exceed the range of the temperature variability.
For the other two nights no cooling outside the temperature variability is observed (not shown).
Due to the enhanced noise of the half-hour sounding many lidar temperature profiles could not be evaluated to
clearly beyond 80 km. This made the calibration by using the GRIPS values impossible. Nevertheless, we used
estimates of the behaviour in the uppermost part of the measurements for a reasonable approach, also looking at
the MOHp profiles. Thus, the results above 70 km are somewhat influenced by the reference to GRIPS. As a



result of this and because of the smaller number of cases that can be averaged at the highest altitudes the
standard deviation diminishes beyond 80 km.
**5. Discussion and Conclusions**
Klanner et al. (2021) demonstrated that with a powerful XeCl laser and the large receiver Raman backscatter
signal accumulated within one hour from atmospheric water can be resolved to more than 25 km during night-
time, allowing us to retrieve useful $H_2O$ mixing ratios to at least 20 km. In this study we examine the
performance of the Rayleigh temperature channel also for one-hour measurements.
The smoothed backscatter signals of the temperature channel can be discerned from the background up to more
than 90 km. Since this corresponds to a small fraction of one count per bin the smoothing must be strong and
currently limits the vertical resolution at high altitudes to 2.9 km (as defined in the 1999 VDI guideline). The
temperatures can be retrieved starting at several kilometres below this. The retrieval in the uppermost part of the
altitude range benefits from the simultaneous OH* measurements at UFS. In cases when the useful range of the
lidar ends below 86 km estimates from extrapolations are used.
The rather systematic positive offset of the lidar temperatures with respect to those from MLS between roughly
40 km and 70 km confirms the expectations from earlier work (Schwartz et al., 2008; Medvedeva et al., 2014;
Wing et al., 2018b; 2021). As mentioned, other satellite-borne sensors, in particular SABER, have shown an
excellent agreement with lidar temperature data above several stations including MOHp.
The agreement of the UFS lidar with the "off" wavelength channel of the MOHp DIAL is better, but on average
the UFS temperatures from roughly 45 km to 70 km also deviate positively. However, several profiles agree
really well which, together with the agreement below 45 km, suggests that there is no fundamental problem with
the retrieval algorithm.
A positive temperature offset means a smaller slope of the air density and, thus, slower decay of the lidar
backscatter signal. A slower decay of the backscatter signal would be typical of known technical artefacts that
will be discussed below. Since the UFS temperatures are higher than those of MOHp this could suggest that the
problems are more on the side of the UFS lidar.
Signal components exhibiting exponential decay are typical artefacts in lidar signal processing. For example,
slow fluorescence decay of optical components, in particular of interference filters, could yield slightly higher
temperatures above a certain threshold altitude. A threshold behaviour around an altitude of about 40 km is,
indeed, observed. However, because of the excellent agreement between the two lidar systems in several cases
we hesitate to assume the presence of fluorescence.
In addition, electronic artefacts must be considered. Wing et al. (2018a) attribute positive T biases above 70 km
to some extent to electronic cross talk or bursts of transient electronic signals and present a method to discard
bad profiles influenced by a shot-by-shot numerical analysis. Such an approach is helpful, but limited to low to
moderate laser repetition rates because of limits in computation speed. In any case, due to sufficient spatial
filtering in our receiver our far-field background count rate is of the order of 0.2 $h^{-1}$ $bin^{-1}$ which means
negligible interference of bursts. In fact, PMT testing with covered photocathode in the detection system verified
even the mostly complete absence of dark counts. The detection system is located far away from sources of
electromagnetic interference in a metallic tower outside the building.
Small exponential wings of the backscatter signal that could give rise to higher temperatures can be caused by
signal-induced nonlinearities of photomultiplier tubes (Bristow et al., 1996). Two effects must be considered,
signal-induced gain change due to overload of the final dynode stages and signal-induced emission of the




photocathode. The first type is avoided by limiting the signal, the second one seems to be suppressed in the PMT
model chosen to a high degree by kinetic-energy filtering (removal) of the artificial low-energy electrons.
Indeed, signal-induced nonlinearities have been found to be absent for the Hamamatsu R7400 detectors under
conservative operating conditions, in particular for photon counting (e.g., Trickl et al., 2020; 2024). Kreipl
(2006) determined the 1/e decay time of an overloaded R7400-03 cathode to less than 30 μs, i.e., much shorter
than the decay time of the backscatter signal of our temperature measurements. These tests had been carried out
at a cathode illumination higher by almost two orders of magnitude. In summary, we can exclude signal-induced
nonlinearities as the reason of the positive temperature deviations.
There is one important difference between the lidar measurements at UFS and MOHp. The measurements at
UFS have taken place before midnight or around midnight the latest. At MOHp, with the exception of the
favourable case in Fig. 6, the data acquisition lasted all night, with measurements times up to more than 12 h
during the cold season. Atmospheric cooling during the second part of the night could serve as an explanation
the observed difference. The GRIPS measurements examined in this study do not reveal reproducible cooling in
the mesopause region with progressing night. The GRIPS temperatures tend to grow with time. At lower
altitudes (e.g.) the examples of Kopp et al. (2015) do not allow us to resolve a temperature change because of the
wave structure superimposed on the temperature profiles. At least our night-time measurement series in
November 2024 demonstrate in these cases that in the almost complete absence of atmospheric cooling the
temperature differences obtained at UFS and MOHp stay strongly below the variability of the 0.5-h UFS lidar
data.
A bias caused by misalignment can be excluded since this would lead to deviations at low altitudes. This kind of
problem has been limited to altitudes below 20 km.
Apart from these issues a range extension to more than 105 km is desirable since this would allow us to retrieve
temperatures with uncertainties of the order of ±5 K up to almost 90 km, which covers the well-mixed part of the
atmosphere (U.S. Standard Atmosphere, 1976). The GRIPS measurement range would then be included in the
useful operating range of the lidar and the temperature obtained from the spectra could be controlled by vertical
sounding. As one can judge from Fig. 1 such a range extension would require a ten times higher laser power.
Since the pulse energy of the current laser is adequate for not exceeding signal levels that ensure linear detection
a much higher pulse repetition rate would be the solution. Diode-laser-pumped Nd:YAG with similar third-
harmonic pulse energies but a repetition rate of more than 300 Hz are meanwhile commercially available. 300
Hz would match the repetition rate of the XeCl laser and water vapour and temperature measurements could be
made simultaneously, with delayed pulses in order to minimize spectral interference. This would mean 15 times
the power of the current transmitter.
In three cases we observed excessive temperature peaks around and above 70 km. Because of the low signal
level at these altitudes we cannot exclude signal outliers which, again, calls for a higher backscatter signal.
However, temperature excursions of the order of 20 K in the mesopause region have been presented in earlier
work and attributed to waves (e.g., Kopp et al., 2015). A clear identification of gravity waves requires to carry
out our time series with a better signal-to-noise ratio.
In addition, an influence of particles or fluorescence from meteorites must be distinguished. One example of a
potential particle layer observed at 54 km with our stratospheric aerosol lidar (Trickl et al., 2013; 2024) is given
in Fig. 10. An identification of the presence of particles cannot be achieved with a simple backscatter lidar. The
particle contribution can be removed by spectral filtering the scattered light from a narrowband frequency-
doubled Nd:YAG laser in iodine (Piironen and Eloranta et al., 1994) . For our system we could think of



additional measurements at the also available 532.2-nm laser emission with an iodine filter as successfully
implemented in our mobile aerosol lidar before (Giehl and Trickl, 2010; Wandinger et al., 2016). The problem of
the light absorption in ozone at 532.2 nm does not matter at high altitudes.
The advantage of symbiotic use of the GRIPS observations at UFS could be strengthened by adding a verticlly
pointing spectrometer. In the near future, the current set-up with two identical spectrometers will be extended by
a third spectrometer that will be aligned for parallel measurements with the laser beam. This, together with less
noiy lidar data, would simplify the interpretation of remaining differences. More detailed studies of the airglow
layer could be made by tuning the fundamental wavelength of the Raman lidar wavelength to 308.15 nm where
OH laser-induced fluorescence can be excited (Brinksma et al., 1998).
After the completion of the upgrading the beginning of routine lidar measurements is planned, as a potential
contribution to NDACC.

**5 Data availability**

Lidar data from the UFS lidar can be obtained from the authors, but are currently flagged as preliminary. GRIPS
temperatures are available at https://zenodo.org/records/15267440. MLS v5 temperature data are available at
https://disc.gsfc.nasa.gov/datasets?page=1&keywords=ML2T_005/. A full list of MLS data can be found at
http://disc.sci.gsfc.nasa.gov/Aura/dataholdings/. The MOHp lidar data can be freely downloaded from the
NDACC data base (https://www-air.larc.nasa.gov/missions/ndacc/data.html).

**6 Author statement**

TT and HV carried out and evaluated the UFS lidar measurements. MB, CS and SW are responsible for the
GRIPS observations, WS for the lidar measurements at MOHp. GN preselected and provided the MLS data. All
authors contributed to preparing this manuscript.

**7 Competing interests**

The authors declare that they have no conflict of interest.

**Acknowledgements**

The authors from IMK-IFU thank Hans Peter Schmid for his support. Lisa Klanner (until 2015) and Matthias
Perfahl strongly contributed to the technical development of the system that recently has included remote
control. We are indepted to the considerable assistance of the UFS staff. The lidar development was funded by
the Bavarian Staatsministerium für Umwelt und Verbraucherschutz (BayStMUV). The GRIPS observations at
UFS have also been supported by BayStMUV in projects including GUDRUN, grant no. 71b-U8729-2003/125-
13; GRIPS3 Back-Up, 2009/40051; BHEA, TLK01U-49580; LUDWIG, TUS01UFS-67093; VoCaS,
TKP01KPB-70581; and AlpEn-DAC, TUS01UFS-72184). Gerald Nedoluha was supported by the NASA Earth
Sciences Division Upper Atmosphere Research Program and by the Office of Naval Research.
The service charges for this open access publication have been covered by a Research Centre of the Helmholtz
Association.

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

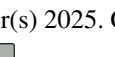


**Figures:**


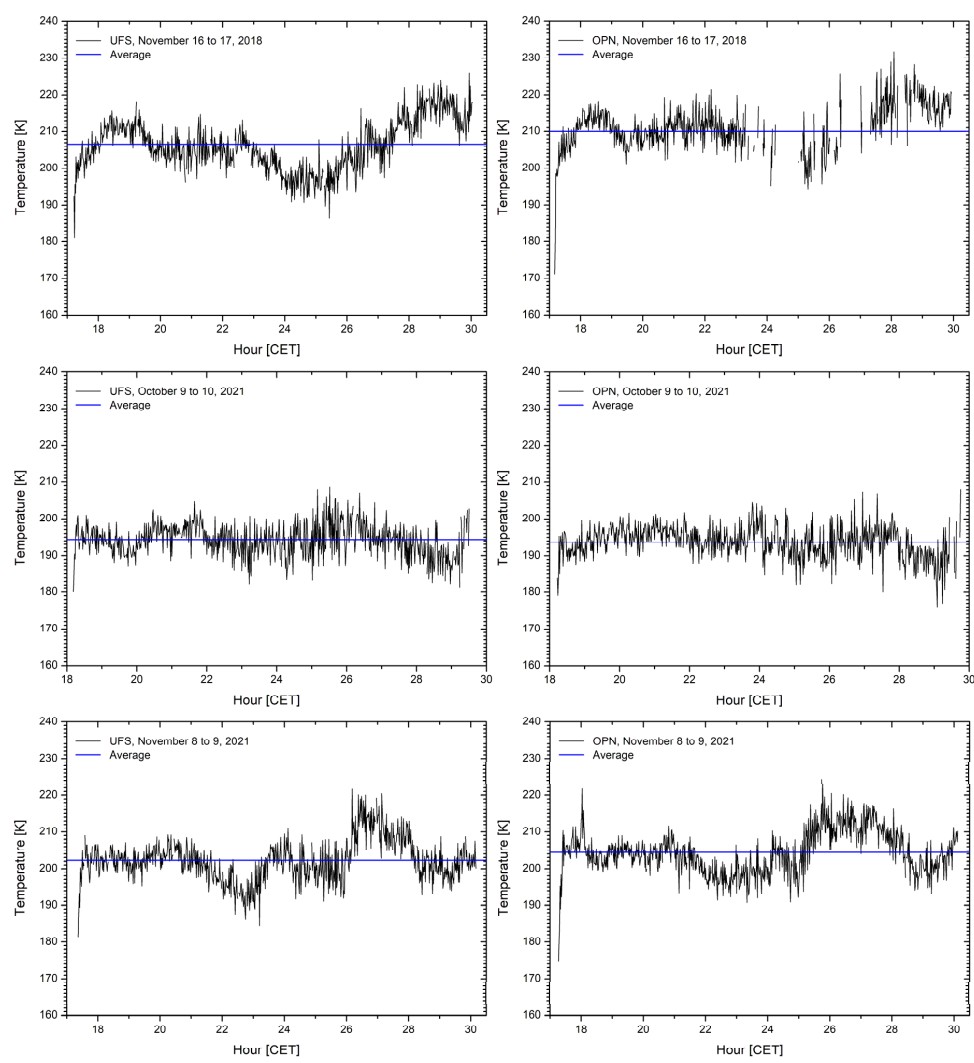

**Fig. 1.** Examples of GRIPS night-time temperature series at UFS (left column) and Oberpfaffenhofen (OPN,
right column) in November 2018, October 2021 and November 2021. There is relatively good agreement for the
first two measurements, but in the third case some differences are seen between the results of the two stations.
Gaps are due to cloud coverage. The uncertainties at the beginning and the end of a given night is high due to
additional daylight.




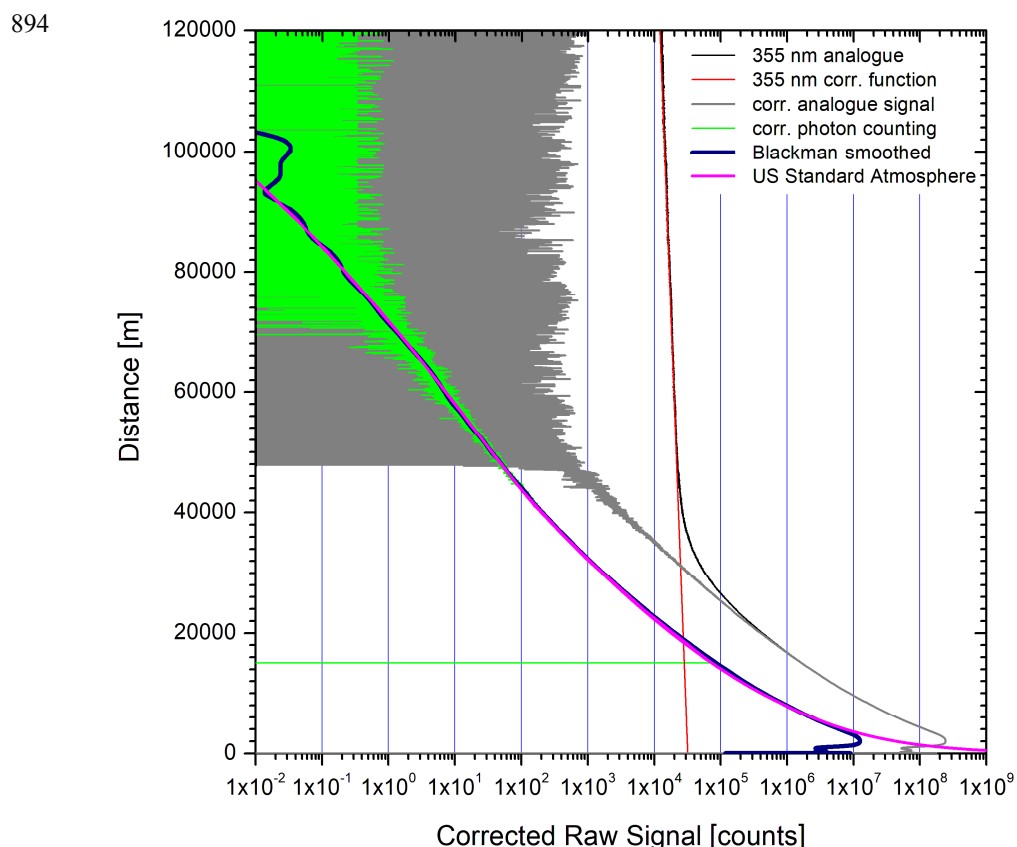

**Fig. 2.** 355-nm backscatter signals accumulated on 9 October 2021 between 23:05 and 24:01 CET (67171 laser
shots) with the transient digitizer and the photon-counting system; the photon-counting signal starts at 15 km due
to a data-acquisition delay (see horizontal green line). Above 100 km just occasionally a single photon was
detected (see scale). The combined signal (after calibrating the analogue signal to counts) from both recorders is
smoothed with a Blackman-type numerical filter with a range quadratically growing with altitude (blue; see
text). For comparison also a synthetic lidar signal simulated with data from the U.S. Standard Atmosphere is
shown.



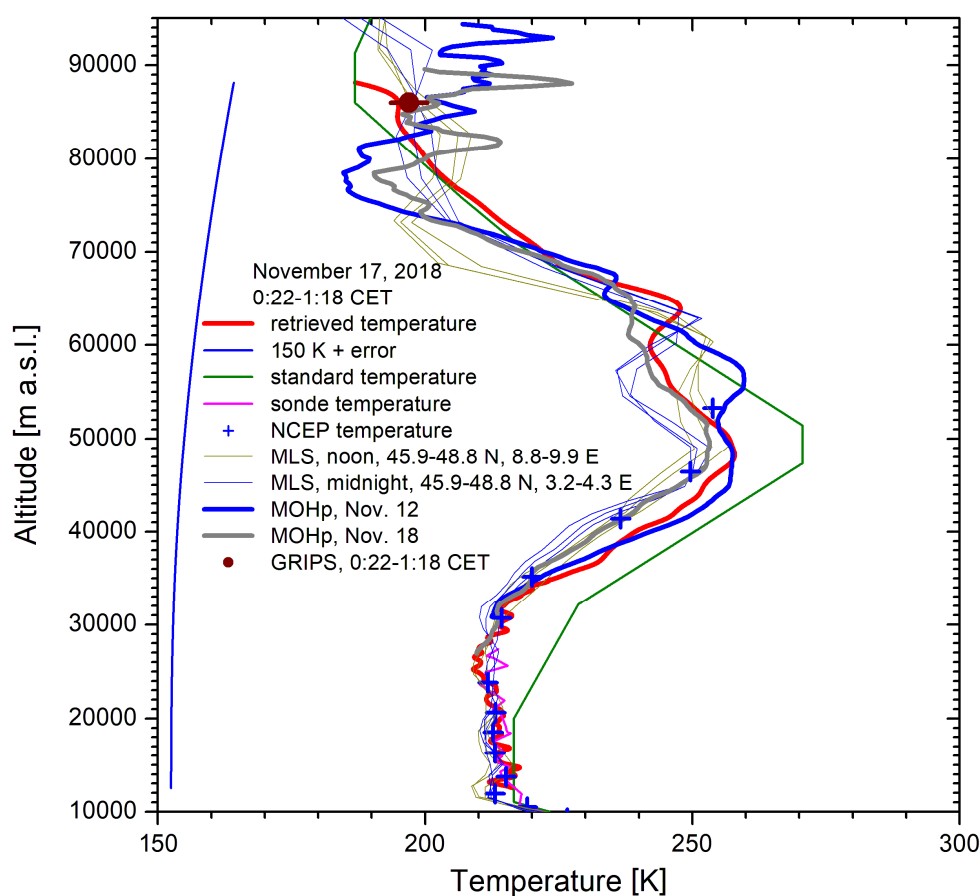

**Fig. 3.** Temperature measurements on 17 November 2018, compared with data and profiles from other sources
(see text).
**Please, print this figure larger than in single-column format to allow visualizing the details.**




**Fig. 4.** Six examples of comparisons of temperature profiles from the UFS lidar with the results of other instruments (for details see legends and text). For a given day, we show both the "midnight" and the "noon" MLS profiles. In the case of Panel (b) the "midnight" MLS profiles are missing and those of the neighbouring two "noon" overpasses are displayed.







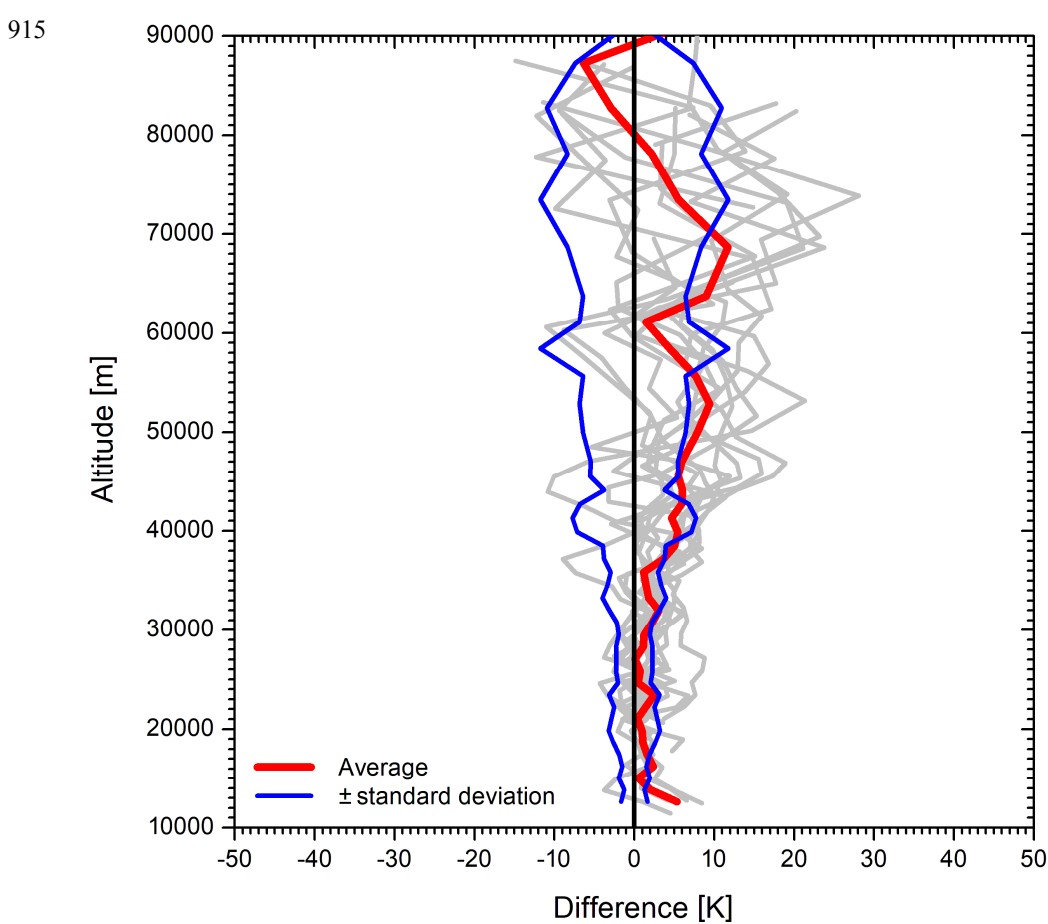

**Fig. 5.** Differences of the 14 temperature profiles of the UFS lidar and the corresponding measurements of MLS
for 47.35º N (grey lines) together with their averages (red line) and standard deviations (blue lines).



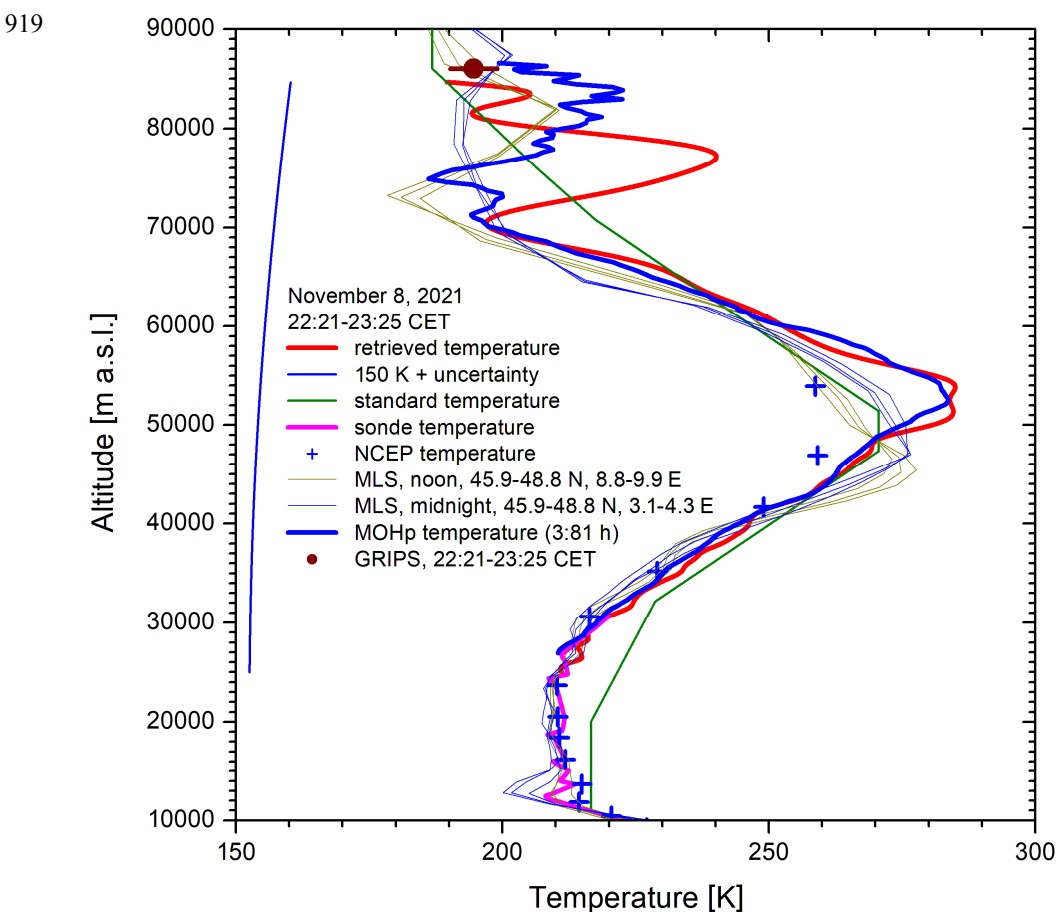

**Fig. 6**. Comparison of the temperature profile from the UFS lidar on 8 November 2021 with the results of other
instruments; here, we observe an almost perfect agreement with the T profile from MOHp up to 70 km, that was
taken during a rather short period between 22:30 CET and 1:49 CET, i.e., almost synchronously to the
measurement at UFS.





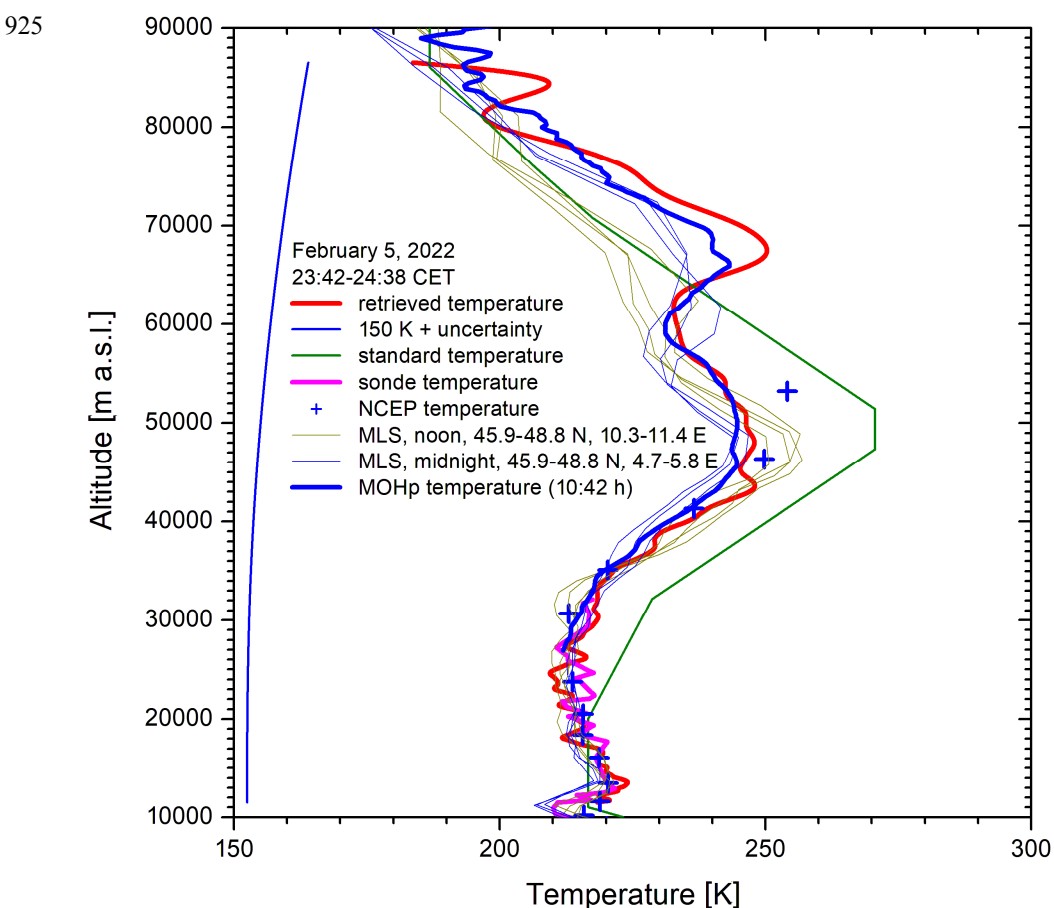

**Fig. 7**. Comparison of the temperature profile from the UFS lidar on 5 February 2022 with the results of other

instruments; here, we observe an almost perfect agreement with the T profile from MOHp, showing rather good

agreement up to 65 km. No OH* measurements are available for that night. The reference temperature was

chosen to match the MOHp temperatures above 77 km (disregarding the ringing).




**Fig. 8**. Two of the four night-time series of the UFS 0.5-h temperature profiles in November 2024, compared with the all-night T profiles obtained at MOHp and the GRIPS average during the lidar measurement periods:

Left panel: 4 November 2024 (start at 21:40 CET) to 5 November 2014 (end at 6:10 CET, beginning of the dawn); groups of three subsequent profiles (M means measurement) are presented share the same colour in order to reduce confusion.

Right panel: 7 November 2014 (0:05 to 6:05 CET); all profiles for measurements 0 to 11 are coloured differently, from black and violet to red.

The standard deviations (STDEV) are shown in blue after adding 100 K.

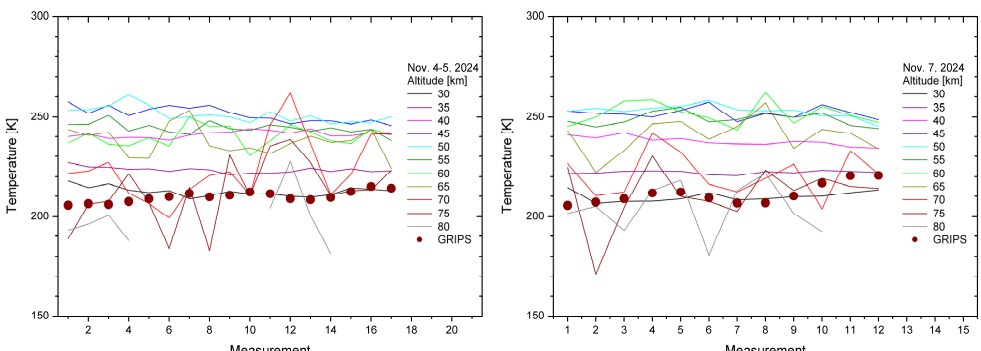

**Fig. 9**. Time series for the two examples shown in Fig. 8 for selected altitudes listed in the legend, together with half-hour averages of the GRIPS temperatures; here, the measurements are labelled from the beginning to the end of the night-time series.






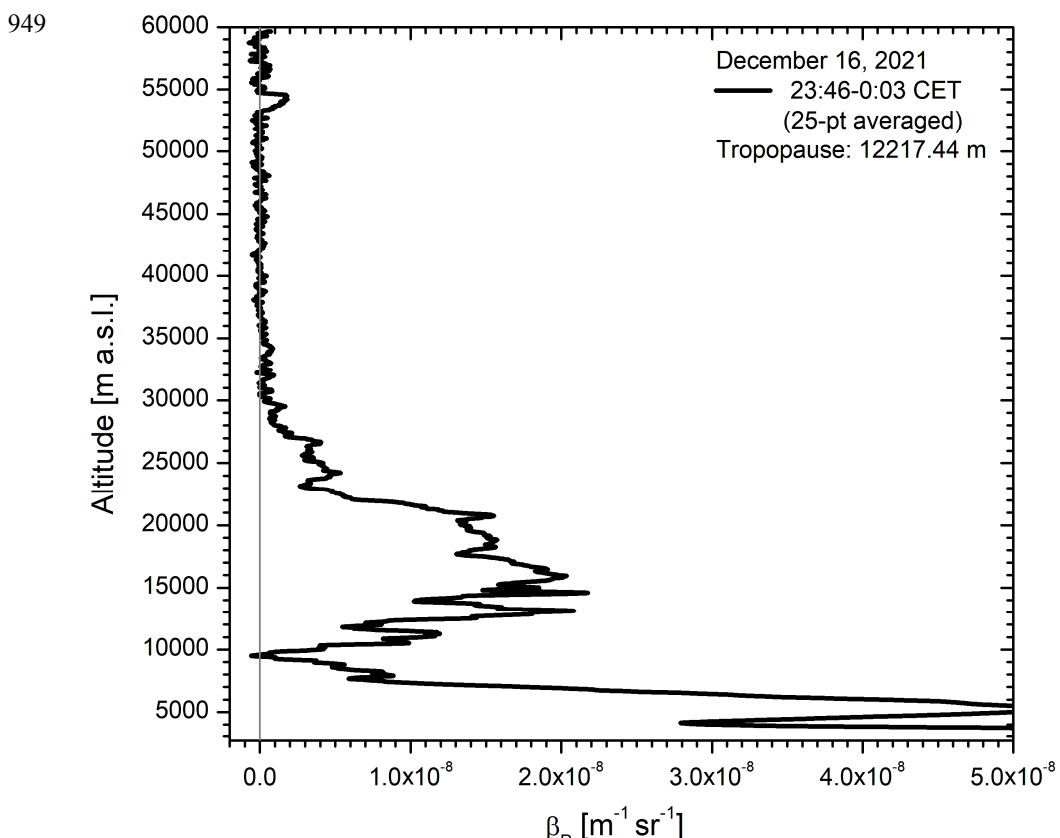

**Fig. 10**. 532.2-nm aerosol backscatter coefficients $\beta_P$ from a night-time measurement with the UFS stratospheric
aerosol lidar showing a strange peak at 54 km.