# Peer review of "Temperature profiles combined from lidar and airglow measurements"

_EGUsphere, 2025_

## Author Comment (AC1)

EGUSPHERE-2025-1952 | Research article

Submitted on 24 Apr 2025

**Temperature profiles combined from lidar and airglow measurements**

Thomas Trickl, Hannes Vogelmann, Michael Bittner, Gerald Nedoluha, Carsten Schmidt, Wolfgang Steinbrecht, and Sabine Wüst

The text of the reviews is given in italics, the replies normal.

**Reply to Review 1:**

This manuscript discusses recent temperature lidar measurements performed by UFS at the Schneefernerhaus high-altitude station near Garmisch-Partenkirchen in Southern Germany. The manuscript is well-written, in proper English, and remains mostly clear. The UFS lidar profiles are compared with profiles measured by other instruments or produced by models, namely NCEP, Aura-MLS, radiosonde and another lidar located 40 km away.

Measuring temperature profiles from this UFS lidar system is the main novel aspect of the paper. The technique itself has been used for several decades, mostly using multi-hour nighttime averages. This new dataset is introduced early in the paper has having the potential to be used for shorter averaging periods such as half hour. After highlighting small but not negligible differences with GRIPS and the MoHP lidar, the authors conclude that additional tests, this time with a co-located (zenith viewing) spectrometer (planned upgrade), will help to fully characterize the UFS lidar temperature measurements, with the ultimate goal to potentially contribute to the ground-based network NDACC.

Out of the previous round of review, the authors have addressed most of the comments I had formulated. From a data user point of view, this new UFS lidar temperature dataset appears to be of very good quality and certainly promising, and for this reason, I recommend the paper to be published after a few more minor revisions/clarifications detailed below:

*Line 365: "the shifted horizontal position of the OH\* measurements"*

For clarity, please use the term "lack of co-location" and refer clearly to the difference in viewing direction of the lidar and GRIPS instruments (the OH\* layer itself is everywhere, not just south of the Central Alps).

"lack of co-location" alone may be confusing since the GRIPS instrument is also located at UFS. I changed the phrase to "the missing co-location caused by the slant-path OH\* measurements (see Sect. 2.3)".

Line 449: "5. At least, the agreement between the two lidar systems is better than that in Fig. 5". Please use a figure similar to Fig. 5 summarizing the difference between the lidars. Such figure would have more value than showing one pmay berofile at a time on multiple figures.

Such a figure, indeed, shows the overall spread of the results and is added. A paragraph was added for the explanation. However, I also see a value in individual examples.

Note that

Line 460-462: "The measurements were carried out with durations of just half an hour"

I think the authors mean here that the temperature profiles were retrieved at half-hour resolution, yet the lidar measurements themselves were continuous throughout the night.

Please re-write this sentence to make this clear (or, if not the case, please clarify what you mean).

Clarified.

Line 562: Typo "verticlly"

Corrected, also another typo in Line 562.

I thoroughly read the manuscript again and made a few additional modifications that are marked with the "track changes" mode.

**Reply to Review 2:**

This paper has done meticulous work in obtaining accurate atmospheric temperature profiles. The temperature reference value of the retrieval is adjusted to match the temperature determined from the OH\* airglow around 86 km by the GRIPS instruments at UFS, or lidar lidar data at MOHp. A detailed comparison was made between the atmospheric temperature profile obtained from UFS LiDAR and NCEP, MLS, radiosonde, and MOHp LiDAR data. A good discussion is provided. There are also some minor modification suggestions:

(1) The 'Fig.1' in section 3.1 should be 'Fig.2'.

I agree that a lidar figure should be the first one. This had initially been the case, but had to be changed because of the citation sequence. Now, a citation of the figure is added in Sect. 2.1 and Fig. 2 becomes Fig. 1 again.

(2) When there is no OH \* airglow data, can you further explain why chose the reference altitude of 77 km using MOHp lidar data (such as Figure 7).

Improved: Not the reference matches! The T profiles of the two lidar systems match on average above 77 km. This required just a slight negative shift with respect to the U.S. Standard T at the reference altitude.

(3) I don't understand the meaning of 'disregarding the ringing' in Figure 7.

Although the T oscillations are clearly visible I removed this phrase. The new explanation is sufficient.